# Analysis of Surface Water Quality in Upstream Province of Vietnamese Mekong Delta Using Multivariate Statistics

**Tran Thi Kim Hong and Nguyen Thanh Giao ***

Department of Environmental Management, College of Environment and Natural Resources, Can Tho University, Can Tho City 900000, Vietnam; ttkhong@ctu.edu.vn
* Correspondence: ntgiao@ctu.edu.vn; Tel.: +84-90-773-9582

**Abstract:** The study employed different statistical approaches to assess surface water quality in the upstream region of the Vietnamese Mekong Delta. The dataset included seven parameters (i.e., temperature, pH, total suspended solids (TSS), five-day biological oxygen demand ($BOD_5$), chemical oxygen demand (COD), ammonium nitrogen ($NH_4^+$-N) and coliform) at seventy-three locations. Cluster analysis (CA) and principal component analysis (PCA) were applied to analyze spatial variations in surface water quality and recognize the important parameters. The findings revealed that surface water quality was deteriorated by organic matters (high $BOD_5$ and COD), nutrients and microorganisms. Particularly, urban areas were found to be more polluted than the other areas. The PCA results indicated that three potential water pollution sources, including industry, urban and tourism, could explain 87.03% of the total variance. Coliform was identified as the leading latent factor that controls surface water quality in the study area. CA grouped the sampling locations into 11 groups, in which the groups of the baseline monitoring sites and large rivers had better water quality. The results indicated a significant impact of anthropogenic activities (especially, urban and tourism practices) in surface water quality degradation. Moreover, CA suggested that the numbers of the sampling sites could be reduced from 73 to 58 locations, lowering 20.54% of the monitoring cost. Thus, the study recommends scrutinizing the current surface water quality monitoring system to be more economic and urgently implementing appropriate solutions to mitigate coliform pollution in the smaller water bodies.

**Keywords:** coliforms; Mekong Delta; multivariate analysis; organic matters; water quality

## 1. Introduction

In the Vietnamese Mekong Delta, the dense river system is considered a strength in economic development and creates favorable conditions for other social activities. However, the impacts of climate change and activities on the mainstream of the Mekong River in recent years have significantly affected water resources in the Mekong Delta. Water resources in the Mekong Delta greatly depend on the upstream of the Mekong River. Therefore, the impacts on the Mekong mainstream have caused many difficulties for the downstream areas. For instance, suspended alluvium content in Chau Doc was nearly 200 g m$^{-3}$ and in Tan Chau about 300 g m$^{-3}$ in flood season in the period 2009–2015 [1]. In the dry season, the suspended sediment concentration in Chau Doc and Tan Chau was about 30–80 g m$^{-3}$. This concentration has decreased from 50–250 g m$^{-3}$ compared to the period 1979–1982 [1]. An increase in the reservoirs of the upstream hydroelectric dam could reduce sediment loads, leading to increased erosion as well as relative sea level rise, and possibly exacerbate water shortages [2]. Agricultural production in the region is also affected by changes in the Mekong River flow. In addition, a large increase in human demands and the discharge of wastes from anthropogenic activities, such as industry, agriculture and aquaculture, have deteriorated this water resource. According to Singh et al. (2005) [3], the main role of rivers is to receive the discharge of domestic, industrial and agricultural wastewater, and they

are also vulnerable to pollution. Therefore, protecting surface water quality has become a matter of the utmost importance to ensure sustainable development. Monitoring at the base line and different activity affected areas is essential for water resource management. This provides insights into spatial changes in surface water quality and promptly corrects negative impacts. In particular, this monitoring is very useful for the provinces in the upper Mekong region (the section that flows through Vietnam), where it may affect the water quality of the downstream provinces. An Giang province is located in the upstream region of the Vietnamese Mekong Delta, with a total natural area of 353,668.02 ha [4]. The province receives all the water from the Mekong River into Vietnam, with two main tributaries, the Tien and Hau rivers [4]. The flow from the Mekong River through two stations Tan Chau and Chau Doc (An Giang province) in October 2020 remained at about 16.000 m s$^{-1}$ [5]. The highest water level in 2000–2020 was recorded to be about 255–506 cm (Tien River) and 235–490 cm (Hau River), decreasing from 7.16–7.4 cm year$^{-1}$ [5]. An Giang is also known as the only province with both banks of the Hau river; consequently determining the water quality of the contiguous provinces (Can Tho city). The hydrological regime depends mainly on the irregular semi-diurnal tide regime of the East Sea, the diurnal tide regime of the West Sea, and the flow from the upstream of the Mekong River [4]. Several previous studies reported that the surface water system in the An Giang province showed signs of pollution [6–8]. However, previous studies only evaluated spatial and temporal water quality variations or multivariate statistical analysis without mentioning different types of impacts on the receiving water bodies.

This study was conducted to evaluate the surface water quality in different impacted water bodies, determining the source and distribution of similar water bodies using multivariate statistical methods. Recently, the multivariate statistical analysis methods, including cluster analysis (CA), discriminant analysis (DA), and principal component analysis (PCA), have been widely applied in water quality assessment [9,10]. These analyses could facilitate processing complex data sets for characterization and water quality evaluation. These methods provide a preliminary explanation of water quality and specifically identify the main influencing factors leading to water quality changes, consequently contributing to recognizing the pollution sources [11–13]. In addition, monitoring too many locations is costly and time-consuming, which can be addressed by grouping locations based on their commonalities in water quality [14,15]. The effectiveness of the multivariate analysis methods in monitoring and evaluating water quality is increasingly recognized. Thus, in this study, the data of 219 samples was collected in 73 locations (both baseline and impact monitoring) and analyzed for 7 parameters (i.e., temperature, pH, total suspended solids (TSS), biological oxygen demand (BOD$_5$), chemical oxygen demand (COD), ammonium nitrogen (NH$_4^+$-N), and total coliform), which were employed for the multivariate statistical approaches. The results of this study could provide more scientific data for the management, monitoring, and identification of pollution sources, as well as the optimization of future water monitoring networks.

## 2. Materials and Methods

### 2.1. Water Collection and Analysis

A total of 219 surface water samples were collected in 2020 at 73 locations, with 3 samplings per year (Figure 1). The study has applied two types of monitoring, baseline and impact monitoring. There are 33 baseline monitoring sites and 40 impact monitoring sites. Specifically, water bodies are mainly affected by flood control, urban, industry, tourism, fishery and agricultural areas. However, agriculture is distributed widely and sporadically throughout the province, which is considered a non-point source. Thus, the impact of this activity was not included in the monitoring program of the province. Impact monitoring sites were divided as follows: flood control area (7 locations, denoted from KS1-KS7), industry (3 locations, CN1-CN3), urban (9 locations, DT1- DT9), fishery (TS1-TS19), tourism (2 locations, DL1-DL2). The detailed information about coordinates, types of monitoring, and characteristics of each monitoring location is presented in Table S1. To

assess water quality, seven water quality parameters were measured, including temperature, pH, total suspended solids (TSS), biological oxygen demand ($BOD_5$), chemical oxygen demand (COD), ammonium nitrogen ($NH_4^+$-N), and total coliform. While temperature and pH were recorded directly in the field by portable meters (ADWA AD11 pH meter and 9142 DO meter) [16], the others were analyzed in the laboratory using standard methods [17], as shown in Table 1. These collected water samples were analyzed at the Center for Monitoring and Engineering of Natural Resources and Environment of An Giang province, with the certificate of VIMCERT 138 (Decision No. 2857/QĐ-BTNMT). TSS was filtered through filter papers and dried to a constant weight at 103–105 °C. The weight increase over the empty dish represents the total solids. BOD was determined by the 5-day BOD test method. The samples were poured into airtight bottles and incubated at room temperature for 5 days. The changes in dissolved oxygen were measured before and after incubation to compute $BOD_5$. To measure COD, the samples were analyzed using the closed reflux and titrimetric method. The samples were refluxed with potassium dichromate and titrated with iron (II) ammonium sulfate.$NH_4^+$-N was analyzed by distillation and colorimetric at 640 nm with a light path of 1 cm or more. Finally, coliforms were determined by the multiple-tube (most probable number) method, which included the detection and counting of coliform bacteria by culture in a liquid medium [18].

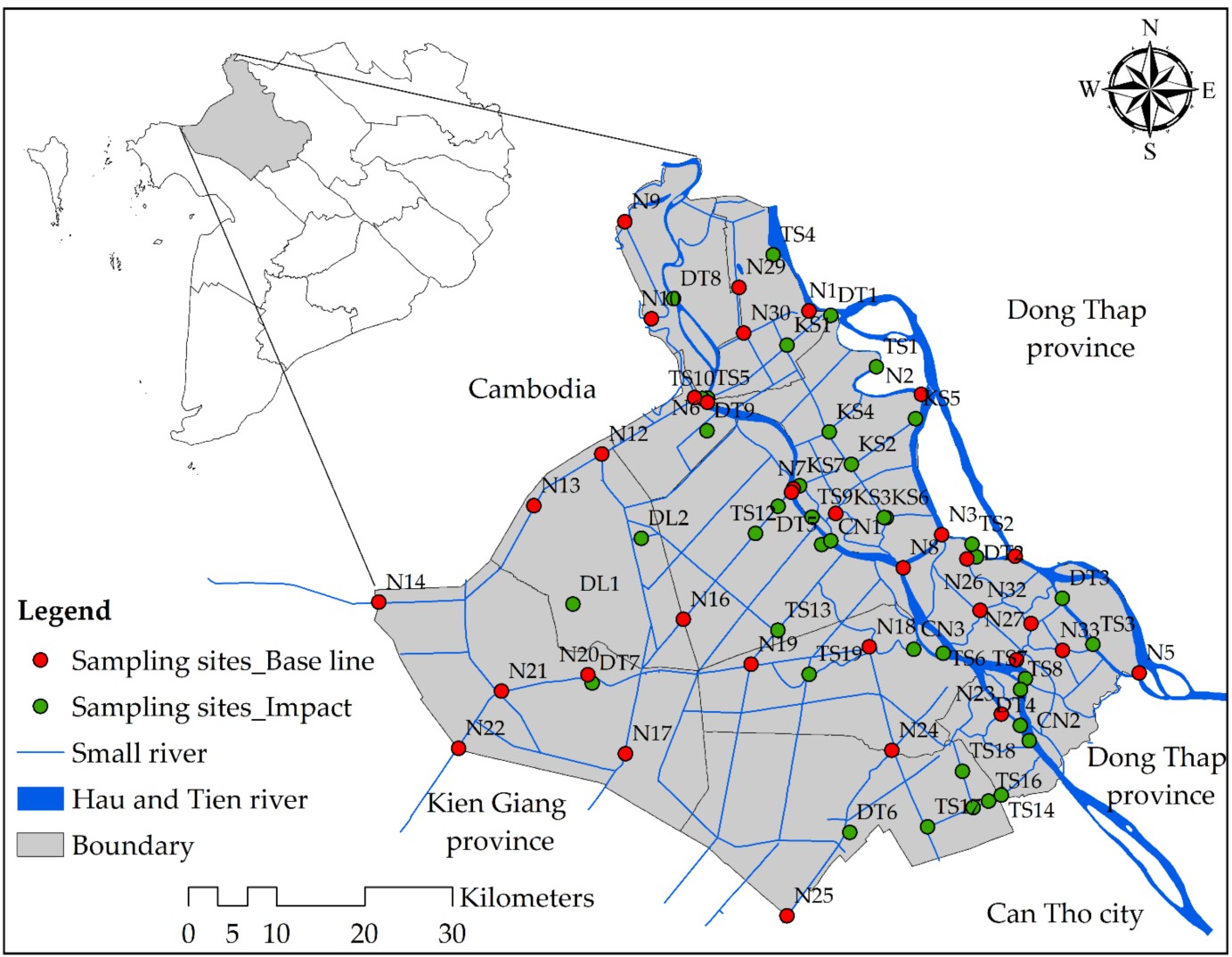

**Figure 1.** Map of the sampling sites in the study area.

**Table 1.** Methods of analysis and allowable limits of surface water quality.

| Parameters | Unit | Analytical Methods [16] | Limit Values [19] | | | |
|---|---|---|---|---|---|---|
| | | | Column A1 | Column A2 | Column B1 | Column B2 |
| Temp. | °C | SMEWW 2550B:2012 | - | - | - | - |
| pH | - | TCVN 6492:2011 | 6–8.5 | 6–8.5 | 5.5–9 | 5.5–9 |
| TSS | mg L$^{-1}$ | SMEWW 2540D:2012 | 20 | 30 | 50 | 100 |
| BOD$_5$ | mg L$^{-1}$ | SMEWW 5210B:2012 | 4 | 6 | 15 | 25 |
| COD | mg L$^{-1}$ | SMEWW 5220C:2012 | 10 | 15 | 30 | 50 |
| NH$_4^+$-N | mg L$^{-1}$ | SMEWW 4500-NH3.B&F:2012 | 0.3 | 0.3 | 0.9 | 0.9 |
| Coliform | MPN 100 mL$^{-1}$ | TCVN 6187-2:1996 | 2500 | 5000 | 7500 | 10,000 |
| Use purposes | - | - | Daily activities, aquatic flora and fauna conservation | Daily activities (must apply appropriate treatment technology) | Irrigation or other uses | Navigation and other purposes with low water quality requirements |

### 2.2. Data Analysis

The average values of the water quality parameters were compared with the standard values in national technical regulations on surface water quality (QCVN 08-MT:2015/BTNMT) (Table 1) [19]. In addition, the study applied one-way ANOVA to determine the statistically significant differences between the different affected water bodies in the study area. The letters a and b were assigned to the values of each parameter in the different water bodies showing statistically significant differences at the 95% confidence level (corresponding to the 0.05 level of significance ($p < 0.05$)). The analysis was performed using SPSS version 20.0 software (IBM Corp., New York, NY, USA).

### 2.2.1. Water Quality Index (WQI)

To assess overall water quality, the water quality index (WQI) is calculated according to the guidance of the Vietnam Environment Administration (VEA) (2019) [20]. However, based on the number of parameters in the study, the WQI calculation formula has been adjusted to Formula (1). The formula and calculation steps are detailed by VEA (2019) [20].

$$\text{WQI} = \frac{\text{WQI}_{\text{pH}}}{100} \times \left[ \frac{1}{3} \sum_{i-1}^{3} \text{WQI}_{\text{II}} \times \text{WQI}_{\text{coliform}} \right]^{\frac{1}{2}} \tag{1}$$

where WQI$_{\text{II}}$ was applied to parameters BOD$_5$, COD and NH$_4^+$-N.

The WQI calculation results were evaluated based on a scale corresponding to different colors and use purposes. There are 6 value WQI ranges to be applied, which are as follows: <10 (heavily polluted, brown), 10–25 (poor, red), 26–50 (bad, orange), 51–57 (moderate, yellow), 76–90 (good, green), and 91–100 (very good, blue).

### 2.2.2. Multivariate Statistics Analysis

Principal component analysis (PCA) was applied to identify the key factors influencing water quality and the potential pollution sources [10,21]. PCA is designed to transform the original variables into new, uncorrelated (axis) variables called principal components (PCs) that are the result of a linear combination of the original variables [14]. The PCs are arranged in descending order of the importance of the parameters in contributing to the interpretation of the data (eigenvalue). According to Liu et al. (2003) [22], the correlation between PCs with water quality variables is divided into strong, moderate and weak, with the absolute value of the weighted correlation coefficient at >0.75, 0.75–0.5 and 0.5–0.3, respectively.

CA was used to analyze the similarities in the composition of the surface water environment at the monitoring locations. The CA results present high homogeneity within the same group and heterogeneity between groups [18]. In the analysis, hierarchical clustering was applied via Ward's method, which used Euclidean distance to measure different similarities between the sites and represented by a dendrogram [13,23–25]. This provides a visual summary of the clustering process. Moreover, the results can help limit the number of monitoring stations. This action was completed by considering the following three conditions: (1) same group, (2) same river, (3) same impact/type monitoring. PCA and CA analyses were performed using Primer 5.2 Windows software (Primer-E Ltd., Plymouth, UK).

## 3. Results

### 3.1. Surface Water Quality Variation

The average values of the physical and biochemical parameters in the upstream region of the Vietnamese Mekong Delta are presented in Table 2. Water quality has been assessed based on the type of water body affected by different activities. The results showed that the water temperatures ranged from $29.62 \pm 0.63$ °C to $29.93 \pm 0.5$ °C and there was no statistically significant difference between the water bodies. However, the temperature measured at the impact monitoring areas tended to be higher than that of the baseline monitoring. The pH value in the study area ranged from $7.15 \pm 0.01$ to $7.22 \pm 0.11$. The pH was relatively stable among the water bodies, and this variation is a statistically insignificant difference ($p > 0.05$). In addition, pH values were recorded within the allowable limit (6–8.5).

**Table 2.** Mean value of surface water quality by water bodies.

| Par. | Base Line | Impact | | | | |
|---|---|---|---|---|---|---|
| | | Urban | Industry | Tourism | Aquaculture | Flood Control |
| Temp. | $29.77 \pm 0.76$ | $29.93 \pm 0.5$ | $29.63 \pm 0.73$ | $29.78 \pm 0.12$ | $29.78 \pm 0.52$ | $29.62 \pm 0.63$ |
| pH | $7.18 \pm 0.08$ | $7.19 \pm 0.14$ | $7.15 \pm 0.01$ | $7.22 \pm 0.11$ | $7.19 \pm 0.06$ | $7.2 \pm 0.09$ |
| TSS | $55.12 \pm 6.8$ | $59.59 \pm 8.82$ | $53.33 \pm 3.59$ | $54.17 \pm 1.65$ | $53.53 \pm 4.8$ | $58.05 \pm 6.21$ |
| COD | $21.14 \pm 6.13$ [b] | $37.22 \pm 18.76$ [a] | $27.92 \pm 4.9$ [ab] | $33.33 \pm 10.37$ [a] | $21.14 \pm 5.75$ [b] | $29.62 \pm 6.47$ [ab] |
| BOD$_5$ | $13.75 \pm 4.04$ [b] | $24.19 \pm 12.21$ [a] | $18.17 \pm 3.17$ [ab] | $21.67 \pm 6.6$ [a] | $13.65 \pm 3.81$ [b] | $19.14 \pm 4.27$ [ab] |
| NH$_4^+$-N | $0.57 \pm 0.42$ [b] | $2.19 \pm 1.74$ [a] | $1.09 \pm 0.37$ [b] | $0.36 \pm 0.04$ [b] | $0.59 \pm 0.46$ [b] | $0.59 \pm 0.24$ [b] |
| Coliform | $16,125 \pm 8587$ [b] | $31,363 \pm 11,476$ [a] | $15,358 \pm 6876$ [b] | $11,067 \pm 2168$ [b] | $16,954 \pm 8150$ [b] | $18,967 \pm 16,645$ [b] |
| Water quality index | 33 | 19 | 28 | 31 | 31 | 38 |

Notes: Different letters indicate statistically significant ($p < 0.05$)

TSS varied from $53.33 \pm 3.59$–$59.59 \pm 8.82$ mg L$^{-1}$ and the difference was not statistically significant between the water bodies (Table 2). The TSS concentrations in the urban-affected water bodies were higher than those in the rest of the water bodies. According to MoNRE (2015) [19], TSS concentration exceeded the permissible limit for column B1 by 1.06–1.20 times.

BOD$_5$ concentration in the study area was from $13.65 \pm 3.81$–$24.19 \pm 12.21$ mg L$^{-1}$. A higher BOD$_5$ concentration was found in the urban water bodies, and the lower was in the aquaculture water bodies. BOD$_5$ exceeded the allowable limit of QCVN 08-MT:2015/BTNMT, column B1 by 1.21–1.61 times [19]. Meanwhile, the baseline and fishery-impacted monitoring areas were still within the limits of column A2. COD concentration in the study area was in the range of $21.14 \pm 6.13$–$37.22 \pm 18.76$ mg L$^{-1}$. The COD and BOD$_5$ concentration in the water bodies affected by urban and tourism showed significant differences compared with the rest of the water bodies ($p < 0.05$) and tended to be higher. Even so, water quality in the urban and tourism areas was not different from the industrial zones and flood control ($p > 0.05$). COD in the urban and tourism areas was higher than the allowable limit of QCVN 08-MT:2015/BTNMT (column B1), which was about 1.11–1.24 times. In the remaining water bodies, COD was only defined beyond the limit of column A2.

For nutrients in the water bodies, $NH_4^+$-N concentration fluctuated from $0.36 \pm 0.04$–$2.19 \pm 1.74$ mg $L^{-1}$, and the highest was found in the urban water bodies. The study also showed that the difference was statistically significant in urban-affected water bodies compared to other water bodies ($p < 0.05$). The concentration of $NH_4^+$-N in the baseline monitoring water bodies, and tourism, aquaculture and flood control areas exceeded the allowable limit of QCVN 08-MT:2015/BTNMT (column A2), about 1.2–1.97 times. Meanwhile, water bodies in the urban and industrial areas were determined to exceed the allowable limit value of column B2 by 1.2–2.4 times. Finally, the coliform density ranged from $11,067 \pm 2168$ to $31,363 \pm 11,476$ MPN 100 mL$^{-1}$, which was higher than the allowable limit of QCVN 08-MT:2015/BTNMT (column B2) by 1.11–3.14 times. The coliform density detected in the urban water bodies was two times higher than in the remaining water bodies. The results of the statistical analysis indicated that coliform density has a significant difference between urban water bodies compared to other water bodies ($p < 0.05$).

The Vietnamese water quality index calculated at the baseline and impact monitoring areas is presented in Table 2. The value of WQI indicated that the water bodies in urban areas were the most polluted, with the WQI value of 19 and rated 4 (red, Table 2) according to the rating scale of VEA (2019) [20]. The remaining water bodies had better water quality ratings, ranging from 28 to 38 (orange, Table 2). However, there was no statistically significant difference in the overall water quality in the water bodies.

*3.2. Principal Component Analysis (PCA)*

As illustrated in Figure 2, four PCs explained 87.03% of the variation in surface water quality in the study area. Three PCs (i.e., PC1, PC2, and PC3) with eigenvalues greater than 1.0 are considered significantly impacting sources [10,21]. However, some water quality indicators in PC4 also contribute significantly to the variation in surface water quality, since their loading factor are high. These were considered to be the main sources of water quality fluctuations in the study area. In addition, the study has identified three sub-sources affecting the water quality of the study area.

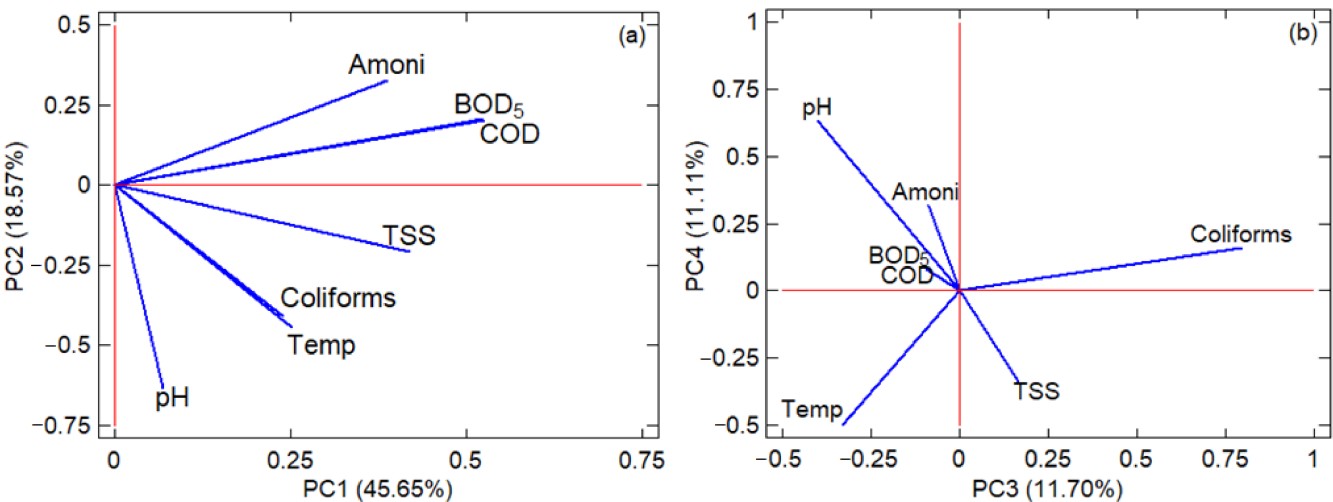

**Figure 2.** Key parameters influencing surface water quality (**a**) PC1 and PC2; (**b**) PC3 and PC4.

PC1 and PC2 could explain 45.65% and 18.57% of the variation in surface water quality, respectively (Figure 2a). PC1 has a weak correlation with TSS (0.42), $NH_4^+$-N (0.39) and a moderate correlation with COD and BOD$_5$ (0.52). PC2 forms a moderate correlation with temperature ($-0.44$) and coliform ($-0.41$), and a weak correlation on pH ($-0.63$) and $NH_4^+$-N (0.33). In addition, Figure 2a also showed positive loadings for BOD$_5$, COD, $NH_4^+$-N and negative loadings for temperature, pH and coliform; this also corresponds to their negative correlation. PC3 and PC4 accounted for 11.70% and 11.11% of the variation in quality in the study area (Figure 2b). It showed moderate and strong loadings with pH

(0.63) and coliform (0.80), respectively. In addition, both PC3 and PC4 showed the limited contribution of BOD$_5$, COD and NH$_4^+$-N, with loadings less than 0.3.

### 3.3. Cluster Analysis (CA)

The CA results indicated that there was a spatial variation in surface water quality, and a total of 73 monitoring sites could be classified into 11 clusters (Figure 3).

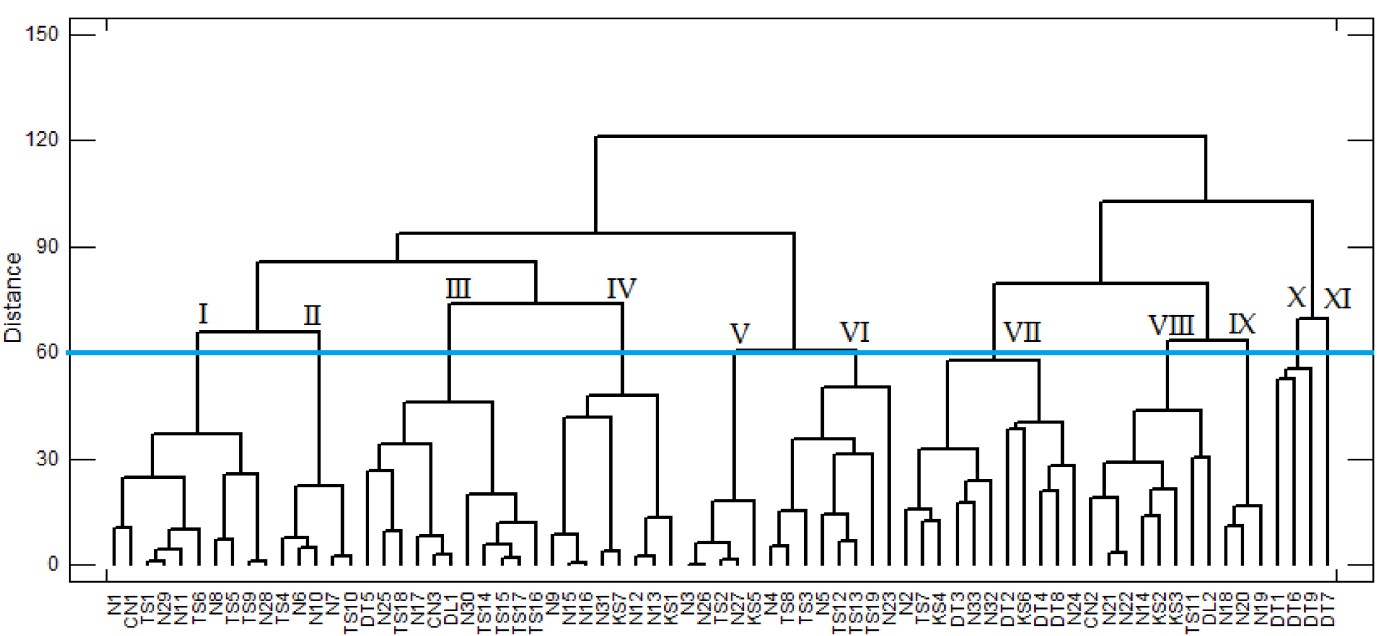

**Figure 3.** Clustering surface water quality in the study area.

Clusters XI included only DT7 (Figures 3 and 4k), which accounted for 1.37% of the surface water variation. Cluster XI belongs to an urban watershed and was identified with the highest levels of TSS, organic and microbiological pollution (Table 3). Meanwhile, Cluster I, III and VII had the highest number of sampling sites with similar water quality. These clusters accounted for 13.70%, 15.07% and 15.07% of the total number of positions, respectively. The positions in Cluster I were mostly identified in the baseline monitoring water bodies on large rivers (Figure 4a). Meanwhile, Cluster III recorded spatially dispersed locations, which belong to water bodies affected by industry, urban, aquaculture and tourism (Figure 4c). Cluster VII was identified in the baseline monitoring area on small rivers and urban areas where samples were collected between the Tien and Hau Rivers (Figure 4g). Next, Clusters IV, VI, and VIII accounted for 10.96% of the sites with similar water quality, corresponding to 8 sites in each cluster. The distribution of locations in Cluster IV was concentrated mainly in the North of An Giang province. The locations belong to two groups of baseline monitoring locations (N9, N12, N13, N15, N16 and N31) and flood control areas (KS1 and KS7). Water quality in Cluster IV was also rated the best (WQI = 54). Similarly, Clusters II and V were formed by five sites, accounting for about 6.85% of total sampling locations. Finally, Clusters IX and X were only formed by three sites per cluster (4.11%). These positions did not have a clear distribution trend (Figure 4i,j. However, the locations in Cluster IX are in the baseline monitoring area and Cluster X is in the water bodies affected by urban areas.

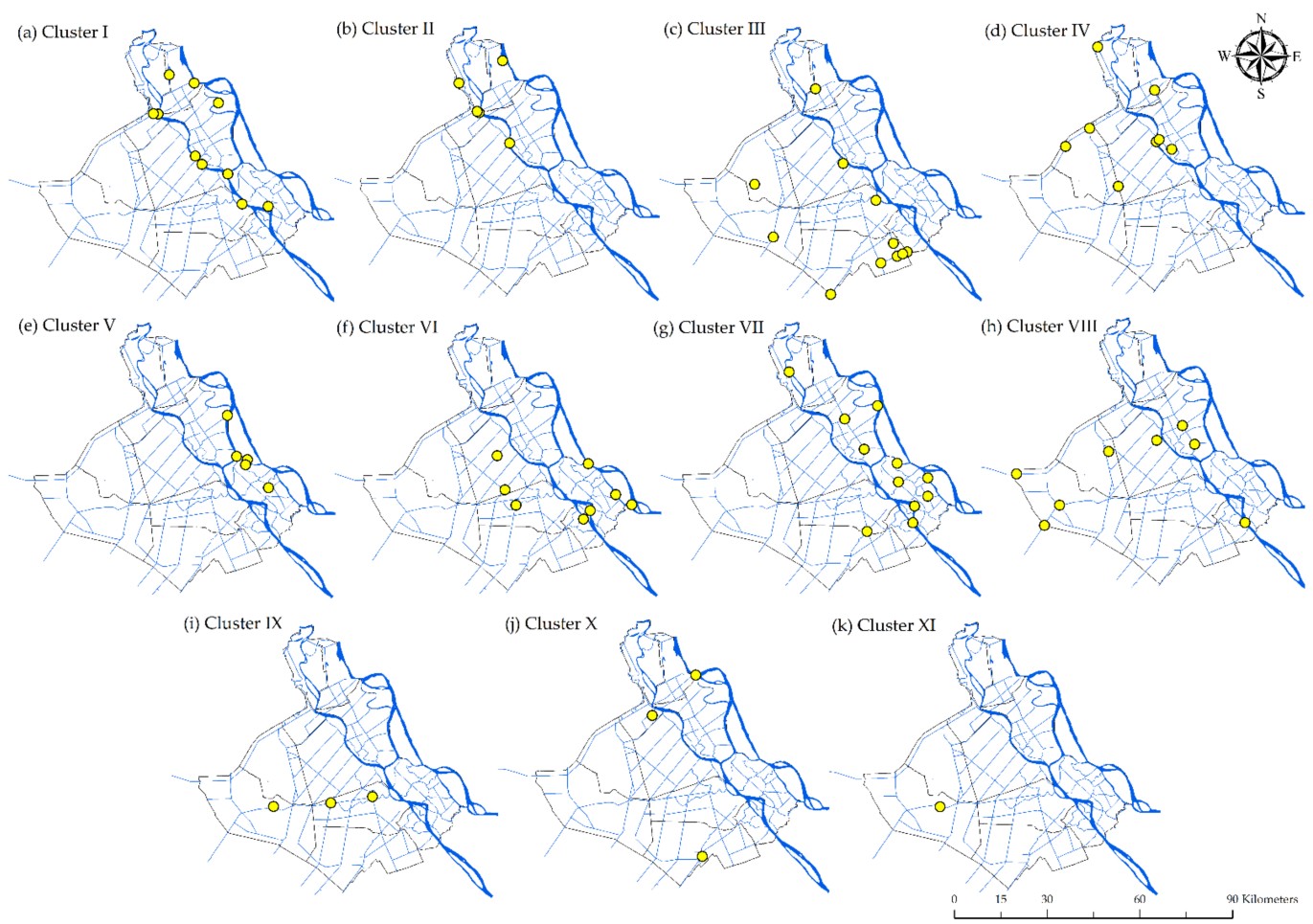

**Figure 4.** The spatial distribution of similar water quality clusters.

**Table 3.** Mean values of water quality parameters obtained from cluster analysis.

| Cluster | I | II | III | IV | V | VI | VII | VIII | IX | X | XI |
|---|---|---|---|---|---|---|---|---|---|---|---|
| Temp. | 29.51 | 29.21 | 29.63 | 28.73 | 30.01 | 30.53 | 29.97 | 29.91 | 30.78 | 30.08 | 30.77 |
| pH | 7.18 | 7.14 | 7.11 | 7.12 | 7.30 | 7.21 | 7.24 | 7.19 | 7.19 | 7.18 | 7.25 |
| TSS | 49.06 | 48.67 | 56.13 | 52.75 | 50.20 | 52.96 | 58.82 | 60.50 | 69.11 | 60.11 | 80.33 |
| COD | 16.94 | 14.87 | 25.14 | 21.79 | 17.14 | 21.04 | 23.48 | 35.10 | 28.89 | 54.33 | 54.67 |
| $BOD_5$ | 10.94 | 9.47 | 16.37 | 14.13 | 11.07 | 13.63 | 15.27 | 22.75 | 18.89 | 35.33 | 35.67 |
| $NH_4^+$-N | 0.44 | 0.24 | 0.88 | 0.40 | 0.40 | 0.69 | 0.78 | 0.96 | 0.56 | 4.47 | 0.65 |
| Coliforms | 11,726 | 26,100 | 15,644 | 7625 | 9453 | 14,150 | 34,603 | 18,488 | 17,378 | 23,178 | 46,000 |
| WQI | 26 | 28 | 21 | 54 | 45 | 23 | 22 | 18 | 22 | 14 | 17 |

## 4. Discussion

### 4.1. Overall Surface Water Assessment

The results revealed that most of the water quality parameters (TSS, $BOD_5$, COD, $NH_4^+$-N and coliform) were over the allowable limits of the national standard (except pH). Several previous studies reported that the water temperature in the Hau and Tien Rivers varied from 27.1 to 32.0 °C [6,26,27] and from 28.6 to 31.5 °C [28], respectively. It can be observed that the measured temperature in this study showed insignificant differences from other regions and was also in the suitable range for the growth of aquatic organisms [29,30]. The pH value in the study was assessed as neutral for different water bodies. In addition, the pH value in the water bodies of An Giang has a more stable fluctuation level than other provinces/cities in the Mekong Delta, such as Hau Giang (6.8–7.1) [31], Ben Tre (6.53–8.02) [32], Dong Thap (7.15–7.36) [33], and Tien Giang (7.2–7.8) [34]. This can be explained by its geographic location in the upstream area, with large water flow, mainly

alluvial soil, less affected by acid sulfate soil through leaching and overflow processes, which helps maintain the pH value as more stable in the water.

TSS contamination of surface water has been deemed one of the major concerns in the Vietnamese Mekong Delta. High TSS concentrations in water bodies can be traced back to soil particle leaching, landslides and other surface runoff. In addition, the flow velocity also partly affects the fluctuation of TSS content in water bodies. This was demonstrated by the highest TSS values detected in the water bodies affected by urban and flood control. However, TSS concentration in the study area tended to be lower than in the Tien river basin (64.0 to 70.8 mg $L^{-1}$) [28]. Suspended solids often exist in an insoluble form, causing water turbidity or dirty and even unsuitable water for domestic uses. In the present study, TSS concentration was only suitable for irrigation and lower water quality purposes. In addition, these substances can block the lights into the water medium, which in turn influences the photosynthesis of plants. Furthermore, TSS is a carrier that helps transport other pollutants, such as pathogenic microorganisms, pesticides, and antibiotics, to different places in the water bodies. Therefore, higher TSS concentration causes an increase in the possibility of human exposure to environmental pollutants and the costs of water treatment.

$BOD_5$ and COD are usually used as indicators of organic waste concentration in water [23,34]. The $BOD_5$ in the impacted water bodies is suitable only for irrigation, navigation, and other equivalent activities. In addition, the $BOD_5$ and COD concentrations in the baseline monitoring water bodies were recorded as suitable for domestic supply water needs. However, appropriate treatment measures were required [19], since carbon compounds (in $BOD_5$ and COD) can combine with chlorine during the disinfection phase to produce hazardous compounds [35]. The significant difference in urban-affected areas may be due to the characteristics of urban wastewater containing high levels of organic compounds. High $BOD_5$ and COD values in the study area are consistent with other water bodies in the Mekong Delta [6,28,31,32,36]. Moreover, the origin of $BOD_5$ can be due to waste from farming, livestock, landfill, domestic activities, and services that have directly discharged untreated waste into the surface medium [37,38].

The enrichment of $NH_4^+$-N is usually derived from nitrogenous waste during the aerobic and anaerobic decomposition of organic matter [25]. It is toxic to aquatic life, especially in alkaline environments [39]. The $NH_4^+$-N values in the water bodies in Hau Giang, Tien Giang, and Dong Thap provinces were 0.27 mg $L^{-1}$ [31], 0.3–0.5 mg $L^{-1}$ [33], and 0.36–0.40 mg $L^{-1}$ [32], respectively. It can be deduced that surface water quality in the water bodies in the Vietnamese Mekong delta was contaminated with nutrients.

According to Ly and Giao (2018) [6], coliform in the surface water of the An Giang province in the period 2009–2016 was over the allowable limit by 2.14–7.02 times. It can be observed that microbial contamination has been a serious problem not only in the study area but also other water bodies in the Mekong Delta [28,31,33,36]. High levels of coliform can either directly or indirectly affect human health via using this water source for domestic purposes, irrigation and aquaculture [40]. As a result, coliform-contaminated water consumption can cause gastrointestinal illness, fever, diarrhea and dehydration [41].

Overall, surface water in different water bodies of the upstream region of the Vietnamese Mekong Delta was contaminated with nutrients, microorganisms, and organic matters. Moreover, this contamination was more serious in urban areas.

### 4.2. Potential Pollution Sources

The results of PCA demonstrated that various potential pollution sources significantly influence the variation in water quality in the study area. Simeonov et al. (2003) [42] and Ma et al. (2020) [43] suggested that the high correlation between PC and organic matter is the result of the impacts of urban and industrial wastewater. Organic matters can be obtained from anthropogenic activities, such as urban, service and tourism, industrial production, and agriculture. According to Barakat et al. (2016) [12], the correlation between PC and pH TSS may be influenced by the physiochemical water properties and the natural weathering process. In addition, Cho et al. (2009) [9] also

suggested that physicochemical pollution from a natural origin largely reflects the ionic characteristics of the water bodies, natural changes in the aquatic medium and the growth status of plankton. Furthermore, PC2 exhibits the influence of microorganisms from human and warm-blooded animal waste. Zhang et al. (2011) [24] also suggested that PC was positively correlated with microorganisms (*F. coli* and *E. coli*), representing fecal contamination. Similar to PC2, PC3 also showed the main influence from human and animal wastes. Moreover, the PCA results indicated that the seven parameters for initial monitoring significantly impacted surface water quality in the An Giang province in 2020. This method identified three main potential water pollution sources of various potential pollution sources. These sources could be a mixture of natural processes and human activities from discharging municipal, industrial, aquaculture and tourism wastewater into surface water systems. Thus, maintaining these seven water quality parameters in the future monitoring program is currently necessary.

*4.3. The Evaluation of Monitoring Locations of Surface Water Quality*

In the cluster analysis, the groups concentrating more sampling locations could mean that there will be several sources of impact and distribution in different water bodies. This has been demonstrated by Cluster III and Cluster VII, where various effects influence surface water quality. In addition, the cluster analysis has also shown the similarity of the locations in the same water bodies affected at relatively close Euclidean distances. In addition, the clusters with baseline monitoring locations in large river areas showed better water quality than those in small rivers (Clusters I, II, IV and V). Furthermore, the number of monitoring positions can be reduced compared to the current monitoring network based on location selection and removal conditions (the locations in the same clusters and water bodies). Specifically, CN1, N8, TS5, TS6 and TS9 in Cluster I belonged to the Hau River. Based on the conditions, TS5, TS6 and TS9 can be potentially eliminated in two of the three positions. Therefore, the study can select three monitoring locations, CN1, N8 and TS5 (TS6 or TS9). Similarly, the locations N1 and TS1 are located on the Tien River. However, these two locations were still kept in the monitoring network because it does not have the same impact (baseline monitoring area (N1) and the impact of fisheries (TS1)). The N11, N28 and N29 locations belonged to Vinh Te, Ong Chuong, and Bay canals, respectively. Therefore, they need to be kept for monitoring. Consequently, Cluster I could be reduced to 8 locations. Likewise, Clusters II (5 locations), Cluster V (5 locations), Cluster VII (11 locations), Cluster IX (3 locations) and Cluster X (3 locations) can be reduced 1 location from each cluster. In addition, Clusters IV, VI and VIII could reduce two locations for each cluster. Meanwhile, Cluster III was determined to decrease from eight positions to five positions. As a result, the monitoring network with a total of 73 locations can be potentially diminished to 58 locations, saving 20.54% of the total monitoring cost [27,32]. Previous studies have relied on only two conditions to exclude sites with similar water quality, namely in the same river and the same group. It can be observed that the application of conditions to remove high similarities in surface water quality positions in the monitoring network in the current study ensures more accuracy than in previous studies. Thus, the number of surface water monitoring sites can be reduced but ensured for surface water quality management.

## 5. Conclusions

The results found that surface water in the upstream region of the Vietnamese Mekong Delta was contaminated with organic matters, nutrients, and microorganisms. A significantly higher water pollution level was found in the water bodies affected by urban and tourism. The overall water quality was only suitable for navigation and other equivalent purposes. The PCA results revealed that three main sources of pollution (industry, urban, and tourism) could explain 87.03% of the variation in water quality. In addition, coliform was identified as the most important parameter affecting the water quality of the study area. CA analysis is effective in grouping the sampling locations based on the similarities

in water quality, which has 11 clusters. The cluster distribution indicated better water quality for the clusters located near large river areas and baseline monitoring (Clusters I, II, IV, and V). Moreover, CA suggested that the total number of monitoring locations could be potentially reduced to 58 locations, saving 20.54% of the monitoring cost. It is recommended that local environmental managers should carefully consider the current findings for the water quality monitoring tasks in the future.

**Supplementary Materials:** The following supporting information can be downloaded at: https://www.mdpi.com/article/10.3390/w14121975/s1, Table S1: Characteristics of the water bodies in the study.

**Author Contributions:** Conceptualization, N.T.G. and T.T.K.H.; methodology, N.T.G. and T.T.K.H.; software, N.T.G. and T.T.K.H.; validation, N.T.G. and T.T.K.H.; formal analysis, N.T.G. and T.T.K.H.; investigation, N.T.G. and T.T.K.H.; resources, N.T.G. and T.T.K.H.; data curation, N.T.G. and T.T.K.H.; writing—original draft preparation, N.T.G. and T.T.K.H.; writing—review and editing, N.T.G. and T.T.K.H.; visualization, N.T.G. and T.T.K.H.; supervision, N.T.G. and T.T.K.H.; project administration, N.T.G. and T.T.K.H. All authors have read and agreed to the published version of the manuscript.

**Funding:** This research received no external funding.

**Institutional Review Board Statement:** Not applicable.

**Informed Consent Statement:** Informed consent was obtained from all subjects involved in the study.

**Data Availability Statement:** Not Applicable.

**Acknowledgments:** The authors would like to thank the Department of Environment and Natural Resources, An Giang province, for the water quality data provision. All the scientific discussion and views in this article are the perceptions of the authors, which do not necessarily reflect the opinions of the data provider.

**Conflicts of Interest:** The authors declare no conflict of interest.

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
