# Peer review of "Analysis of Surface Water Quality in Upstream Province of Vietnamese Mekong Delta Using Multivariate Statistics"

_water, doi:10.3390/w14121975_

Round 1

Reviewer 1 Report

The authors have satisfactorily addressed the raised concerns.

Author Response

Dear Reviewer,

The authors would like to thank four your comments that signficantly help in improving the quality of the revised manuscript.

Kind regards,

Authors

Reviewer 2 Report

Dear authors,

after revising your MS and dealing with the reviewers' comments and suggestions, your paper looks better and improved, but still needs some polishing prior to publication in Water.

Comments:

1. Introduction: It would be beneficial to include the study design along with the parameters and number of sampling locations to better describe the study.

2. There must be no confusion regarding the parameters analysed - is it ammonium ion, ammonia, or ammonium nitrogen? From the chemical formula, it looks like ammonium nitrogen, but you describe it as ammonium (L11) and ammonia (L96). Please be consistent.

3. "BOD" should be replaced with "BOD5" throughout the text (L10, 95, 158-163, 201, 203, 209, 241, 269-279,  Tables 1-3 and Figures 2a, 2b.

4. Introduction, L14: Please describe "organic matters" or chose a different term.

5 Please add references for the used statistical data - L35, L38, L53, L56 and L57.

6. Introduction, L75: Please add the aim of the study before explaining the methods and the results in brief.

7. Materials and methods, L97: Describe "hand-held" or chose a different term.

8. Materials and methods, L98: Describe the used methods in brief. A reference to the APHA methods and their abbreviations (Table 1) is not sufficient. The apparatus used along with information about manufacturer, country, town, and model should be included. Information about the laboratory used (accreditation under ISO/IEC 17025 should be stated) and Quality control measures - uses of certified reference material(s), recoveries, uncertainty, etc. should be included.

Author Response

Dear Reviewer,

The authors would like to thank for your comments that significantly helps in improving the quality of the final manuscript. Please find the revised MS and responses in the attachment.

Kind regards,

Authors

This manuscript is a resubmission of an earlier submission. The following is a list of the peer review reports and author responses from that submission.

Round 1

Reviewer 1 Report

In this study, Hong and Giao assessed surface water quality in the upstream region of the Vietnamese Mekong Delta using Cluster Analysis (CA) and Principal Component Analysis (PCA) as statistical tools. Seven water parameters were identified and analyzed at 73 locations. The monitored water quality parameters were assessed against the national technical regulations. The analysis and assessment concluded that surface water in the study area was polluted with different contaminants, including organic matters, nutrients, and microorganisms. Most tested parameters had far exceeded the acceptable standards. PCA revealed that three primary pollution sources (municipal, industrial, aquacultural, and tourism WW) could explain 80% of water quality variations. At the same time, CA suggested a possible remarkable reduction in the number of monitoring locations.

Overall, the study is good and informative, and the manuscript is well-organized with appropriate methods and discussions. The introduction is concise, compact, and provides enough background. There are no major technical issues; however, some minor comments should be addressed before accepting the paper's publication in Water. Following are my comments:

  1. Cost is not necessarily directly proportional to the number of locations. Thus, the reported reduction in cost due to reduced sampling sites could be misleading and should be avoided unless adequately explained.

  1. Section 2: A small map of the area under study identifying the main mentioned attributes under section 2 might better explain the location and its importance. It can be merged into Figure 1.

  1. Table 1: according to the cited standard, since they are discussed in the text, all columns of A1, A2, B1, and B2 should be included, and their allowed use purposes should be defined in the table caption.

  1. Figure 2: Fonts are tiny and difficult to read. Specifically, please increase the font size of the y-axis and the dots/asterisks. Also, make sure the colors are consistent between what the figure shows and what is written in the text.

  1. There is far too much reporting of water quality parameters from previous studies in the discussion section, especially in section 4.1. While reporting and comparing to prior studies is essential, the focus of this section should be to "discuss" and explain rather than report.

  1. Section 4.1. please discuss the possible origin/sources of TSS found in water samples.

  1. More details should be discussed on how and based on what the sampling locations can be reduced. For instance, why can the sites within Cluster 6 be reduced from 34 to 23 (but not to any other number)?

  1. Some minor comments:
  • Line 117: the middle line of the boxplot in the figure is green, not blue.
  • Please pay attention to the proper numbering of sections. Data analysis should be 3.2. Results should be 4. (and subsections of 4).
  • Line 176 and 182; did you mean 2e (instead of 2f) and 2f (instead of 2g)?
  • Please be consistent in reporting your values. For example, in section 4.1., some values were reported with SD while others were without.

Reviewer 2 Report

This paper is well written. I only have some minor suggestions.

  1. Line 26 - 38. This information is too much and not very related to water quality. I suggest the authors to simplify it into two or three sentences.
  2. Section 2 should be combined in Introduction.
  3. about the method, the authors should point out how many samples were collected in total and what was the sampling frequency. 
  4. Figure 1 need to be improved. The resolution is not high and hard to read.

Reviewer 3 Report

The authors studied surface water quality Mekong Delta using PCA, Cluster analysis and suggested a reduction in sampling stations based on statistical analysis. My observations are as under:

Spatial and temporal variation in water quality data is not clear. Like axis for Fig. 2 are not clear (Variation of surface water quality in the study area: (a) pH, (b) TSS, (c) BOD, (d) COD, (e) 158 NH4+-N, (f) Coliforms).

Figure 1 (Map of the sampling sites) is not clear. Also, draw a line diagram and clearly mention station type, and station name

In Table 3 (Key parameters influencing surface water quality)- It is always better if you take eigen value>1 or 80% variance explained or both. I feel 3 or 4 principal components may give a better picture. Also, in your analysis, many factors are confounded with different PCs like pH is PC3 and PC4. Thus, its justification becomes difficult. Try Varimax rotation which may yield better results.

Try to apply Water Quality Index (WQI) to check changes in water quality.

Do mention which station is Base line, Impact or Trend station. Refer to Water Quality Monitoring - A Practical Guide to the Design and Implementation of Freshwater Quality Studies and Monitoring Programmes (1996 UNEP/WHO)

Section 3.1. Water collection and analysis. Better make a table for stations. The last portion of this section is too rudimentary.

What about flow data. This is the most significant parameter.

What is the source of data/sampling dates? Do mention the number of samples (n) in Table-2